# A Study on Accessing Linguistic Information in Pre-Trained Language Models by Using Prompts

**Marion Di Marco**[1] and **Katharina Hämmerl**[1,2] and **Alexander Fraser**[1,2]

[1]Center for Information and Language Processing, LMU Munich, Germany

[2]Munich Center for Machine Learning, Germany

{dimarco,haemmerl,fraser}@cis.lmu.de

## Abstract

We study whether linguistic information in pre-trained multilingual language models can be accessed by human language: So far, there is no easy method to directly obtain linguistic information and gain insights into the linguistic principles encoded in such models. We use the technique of prompting and formulate linguistic tasks to test the LM's access to explicit grammatical principles and study how effective this method is at providing access to linguistic features. Our experiments on German, Icelandic and Spanish show that some linguistic properties can in fact be accessed through prompting, whereas others are harder to capture.

## 1 Introduction

With pre-trained language models becoming ever larger and continuously better at solving a variety of tasks, there is a growing interest in understanding how they represent knowledge, particularly linguistic information. While pre-trained language models carry out impressive feats of linguistic generalization, there is no simple way to directly access the linguistic knowledge encoded in them. Easy access to linguistic information, and — in a more long-term vision — the ability to modify parts of the linguistic representation, would be a powerful instrument. For instance, a linguist without expert knowledge in machine learning could use linguistic querying to gain insights into a model's linguistic properties and to pinpoint problems with its linguistic representation. Similarly, for a language learner, interacting with a language model in the form of textual instructions could provide interesting new learning opportunities.

In this work, we focus on accessing linguistic information through natural language queries. We propose using prompting as a simple way to test the LM's access to explicit grammar principles. We study on German, Icelandic and Spanish how effective prompting is for querying linguistic informa-

tion, and what type of information can be obtained, versus what is out of reach for the method. There has been limited work on prompting English models for English linguistic information, while our focus is on other languages. We are also interested in prompting in one language and obtaining linguistic information in another, such as a language learner or linguist might do. This will be also of interest for working with linguistic information in low-resource languages, both when looking only at a low-resource language individually, or in combination with another (high-resource) language, such as the pair of Spanish and Mayan languages.

In our experiments, we formulate question prompts for different linguistic tasks, for example to "ask" for the grammatical case of a word in a given context sentence. The tasks are designed as classification tasks, where the labels correspond to linguistic properties (e.g., *singular, plural* for the feature *number*). Our experiments address standard morphological features, such as *tense, case, number, gender*, but also syntactic information, such as the distinction between subject and object, and the distinction between active and passive voice.

Our findings indicate that features corresponding to concrete and locally defined properties like *number* of nouns are easier to obtain than more abstract features like the grammatical *case* of nouns. They also support the hypothesis that abstract grammatical terms (e.g., *dative case*) are not always well-suited for formulating prompts or answers as they are not necessarily well-covered in the grammatical sense in general-language data.

We present this analysis as some first steps in evaluating how relevant linguistic properties of a language can be made accessible through human language queries. Possible next steps can be to focus on low-resource languages, as well as identifying problems in representing linguistic information and strategies to solve such problems.

## 2 Related Work

There is strong interest in how linguistic information is represented in pre-trained LMs: Liu et al. (2019) is a well-known probing study transferring linear probing techniques to contextual models. Shapiro et al. (2021) study multilabel probing to assess the morphosyntactic representations of multilingual word embeddings. Stanczak et al. (2022) investigate whether the same subsets of neurons encode universal morpho-syntactic information and therefore enable cross-lingual generalization. Lasri et al. (2022) study how BERT encodes grammatical number in a number-agreement task. Hewitt and Manning (2019) by contrast focus on *structural probes* to find out whether BERT and similar models encode parse trees according to their theory. Warstadt et al. (2020) provide a challenge set using acceptability contrasts to evaluate the morphosyntactic and semantic knowledge of LMs, testing for a range of phenomena. Liu et al. (2021) is a probing study of English RoBERTa across time. The authors test a range of probes and downstream tasks at dozens of checkpoints. Li et al. (2022) argue that prompting acts as a model-free probe, thus eliminating the distinction between what the model knows and what the probe learns. They compare prompting to linear regression and MLP probing. Zhang et al. (2022) also probe GPT-3 using prompts, specifically on tense and number features. Blevins et al. (2023) prompt GPT models to append POS tags to each word in a given sentence. However, most of the above works focus on English-only settings. We work with multilingual models and for the presented experiments, use mixed-language prompts, where the question is formulated in English to obtain information about a another language, in which the target word and the context sentence are given.

## 3 Methodology: PET

We apply *Pattern-Exploiting Training* (PET: Schick and Schütze (2021a), Schick and Schütze (2021b), Schick and Schütze (2022)).[1] PET is a semi-supervised training procedure which combines textual instructions with fine-tuning based on labeled examples. Its key idea is providing textual instructions for the task (prompts) as cloze-style phrases for a masked language model (Devlin et al., 2019), where the mask token is then substituted by one of

---

[1] We use the implementation in https://github.com/timoschick/pet

| Sentence given as context: | Label: |
|---|---|
| Die **Regelung** ist stark umstritten. | SG |
| *The **regulation** is highly controversial.* | *gloss* |
| Sentence. " Regelung " is _MASK_ . | |
| Sentence. The number of " Regelung " is _MASK_ . | |
| Sentence. Number of " Regelung " : _MASK_ . | |
| Verbalizer mapping: PL → plural, SG → singular | |

Table 1: Prompts for the nominal feature *number*.

a set of possible output tokens. A *verbalizer* maps the class labels to natural-language outputs, and completions are sampled exclusively from this set of output options. One example from the original paper is predicting restaurant ratings given reviews, where the verbalizer maps "terrible" to a 1-star-rating and "great" to 5 stars. In our experiments, the output classes typically correspond to the values of grammatical features to be predicted, for example *plural, singular* for the feature *number*.

Typically, the training of PET relies on several *patterns*, namely variations of the prompts. In a first step, individual models are fine-tuned on a small training set for each pattern. The resulting models are then used to label a larger set of unlabeled examples with soft labels as training data for a final classifier with a regular sequence classification head, similar to knowledge distillation.

PET has two advantages for our experiments: It is designed to work well with small sets of training data, which might be useful for tasks with only few annotated data available; and it combines several prompt formulations for the same task, which alleviates the problem of finding the "right" prompt.

## 4 Prompt Design and Data

To train PET, we need linguistically annotated data to derive prompts and the respective labels which are then mapped to natural-language words as candidates for the mask position in the LM.

**Prompts and Labels** For the formulation of prompts, we first present a sentence as context, and then variations of statements like "*the feature of W is _MASK_*", with *W* being a word in the sentence. Table 1 shows example prompts and the label vocabulary for the feature *number*.

We mix the language of the prompt (English) and the languages of the queried feature (German, Icelandic, Spanish). Even though multilingual language models are robust to mixed-language input, there are some considerations: The terminology used in the prompts, namely grammatical features

and their values, might be insufficiently covered in the underlying training data of the language model in general, and potentially even more so for English labels referring to another language's grammar. For some terms, there might be a domain shift (grammatical gender vs. gender in the societal sense).

With the success of the method hinging on the coverage of the relevant terminology (in English or otherwise), we also experiment with substituting grammatical terms with general-language discriminative context of the respective target languages (cf. "variants" in Section 5.1).

**Data** Our datasets[2] stem from the Universal Dependency Treebank (Nivre et al., 2020), which contains high-quality data for many languages. Our main focus is on German, which has a complex nominal morphology with many syncretic forms making it often impossible to derive a word's features without context. In addition, we choose Icelandic, which has a similar feature distribution (three gender values, four case values), but is a lower-resource language compared to German. On the other hand, Spanish has a comparatively simple nominal morphology (two gender values, no noun cases), but a more complex verbal morphology, largely without syncretic forms.

For the experiments on morphological features, we select the target words (nouns and finite verbs) such that that each combination of lemma and features occurs only once, and that there is no overlap between test and training data. The target word can only occur once in the sentence, and we restrict sentence length to 50 words. Table 8 in the Appendix shows the amount of training data per language.

Table 3 provides an overview of the predicted features and the respective labels. For the feature *Verb Tense*, we only look at finite verbs, but do not consider composed tenses (e.g. *machte* vs. *habe gemacht* (*made* vs. *have made*)). For Spanish, the corpus annotation differentiates between two different forms of past tense (*Past: preterite tense*) and *Imp: imperfect tense*). To limit this prediction task to the more general prediction of *past - present - future*, the two Spanish past tenses were mapped into one "Past" label. Furthermore, Spanish expresses future tense morphologically, whereas German and Icelandic use an auxiliary: *(yo) andaré* vs. *ich werde gehen* (*I will go*).

The different feature values do not always oc-

---

[2]The data sets can be found at https://github.com/mariondimarco/morphPET_dataset/

|  |  | DE | | IS | | ES | |
|---|---|---|---|---|---|---|---|
|  |  | train | test | train | test | train | test |
| **Tense** | past | 27 | 380 | 14 | 80 | 29 | 448 |
|  | pres | 73 | 446 | 86 | 117 | 47 | 401 |
|  | fut | – | – | – | – | 24 | 83 |
| **Number** | sing | 74 | 563 | 64 | 145 | 58 | 608 |
| **Nouns** | plur | 26 | 263 | 36 | 52 | 42 | 324 |
| **Number** | sing | 74 | 514 | 60 | 137 | 54 | 615 |
| **Verbs** | plur | 26 | 312 | 40 | 60 | 46 | 317 |
| **Gender** | masc | 36 | 338 | 31 | 71 | 58 | 489 |
|  | fem | 40 | 342 | 41 | 64 | 42 | 443 |
|  | neut | 24 | 146 | 28 | 62 | – | – |
| **Case** | acc | 41 | 254 | 33 | 70 | – | – |
|  | dat | 25 | 228 | 27 | 57 | – | – |
|  | gen | 5 | 85 | 14 | 11 | – | – |
|  | nom | 29 | 259 | 26 | 59 | – | – |

Table 2: Label distribution per feature and language.

|  | DE | ES | IS |
|---|---|---|---|
| **Tense** | past | past | past |
|  | present | present | present |
|  | – | future | – |
| **Number** | singular | singular | singular |
|  | plural | plural | plural |
| **Gender** | masculine | masculine | masculine |
|  | feminine | feminine | feminine |
|  | neuter | – | neuter |
| **Case** | accusative | – | accusative |
|  | dative | – | dative |
|  | genitive | – | genitive |
|  | nominative | – | nominative |

Table 3: Labels for the morphological features.

cur with proportional frequency, leading in some cases to very unbalanced data sets (see Table 2). We decided not to balance the data sets, but to use the frequencies as they occurred on randomly chosen words to preserve the approximate proportions between the different features.

## 5 Experiments

For the experiments on morphological features, we use finite verbs and nouns as words *W*. While the nominal features *number, gender* and *case* are also present for adjectives and determiners, we chose to only look at the phrase head for the sake of simplicity, assuming that if the feature of the head is known, it is also known for the rest of the phrase.

In our tables, we give the accuracy of the final distilled model for *bert-base-multilingual-cased*, and *xlm-roberta-large* (Devlin et al. (2019), Conneau et al. (2020)). We report the average over

two runs with different seeds,[3] using the default parameters of PET.

For each set of experiments in this section, we use 100 training examples, 1000 (DE/ES) or 500 (IS) examples to be labeled for the final model (cf. Table 8). Using the same sets of prompts for all languages, formulated in English, we assume that the different prompts do not do equally across the languages. We use the *dev* set to select the 3 best out of 5 initially given patterns.

## 5.1 Morphological Features: Nouns and Verbs

Table 4 gives an overview of the experiments: For *number*[4] and *tense*, the predictions are quite high (with the exception of Icelandic *tense*), even when considering that the number of output classes is not balanced (cf. Table 2). This is contrasted by the performance for *gender* and particularly *case*; increasing the amount of labeled training examples only leads to a moderate improvement in German.

Case is complex and context-dependent; it often corresponds to the syntactic function of a subcategorized noun, and thus requires a general understanding of the entire sentence. Gender is a local feature innate to a noun, with mostly arbitrary values that do not correspond to real-world properties and are inconsistent between languages. The features tense and number are more tangible in the sense that the terms used for their description are meaningful and also prevalent in general language.

Particularly for German, the high number of syncretic forms makes the learning task even more difficult. While there can be context keys, such as determiners and inflectional suffixes within the noun phrase, this is, however, limited to certain contexts and inflectional morphemes are often not directly accessible due to fusional morphology and inconsistent subword segmentation.

The results of Icelandic are notably worse than those of the other languages. One possible factor might be that the proportion of Icelandic in the PLM training data is considerably smaller than for German or Spanish; furthermore, our data set for Icelandic also comprises less training data.

Another issue is the unbalancedness of some of the data sets; as a result, we observed that in some

| finite VERBS | | DE | IS | ES |
|---|---|---|---|---|
| **Tense** | mbert | 0.841 | 0.594 | 0.932 |
| | xlmr-large | 0.965 | 0.594 | 0.990 |
| **Number** | mbert | 0.792 | 0.790 | 0.995 |
| | xlmr-large | $0.725^D$ | 0.954 | 1.00 |

| NOUNS | | DE | IS | ES |
|---|---|---|---|---|
| **Number** | mbert | 0.914 | $0.762^D$ | 0.999 |
| | xlmr-large | 0.942 | $0.843^D$ | 0.998 |
| **Gender** | mbert | 0.414 | 0.325 | 0.483 |
| | xlmr-large | 0.765 | $0.544^D$ | 0.993 |

| Case | DE-100 | DE-250 | DE-500 | IS |
|---|---|---|---|---|
| mbert | 0.328 | 0.367 | 0.444 | 0.363 |
| xlmr-large | 0.313 | 0.334 | 0.377 | 0.307 |

Table 4: Accuracy for tense, number, gender and case. D means the two runs differed by > 0.2.

| Gender | DE-100 | DE-250 | DE-500 | IS |
|---|---|---|---|---|
| mbert | 0.815 | 0.838 | 0.851 | 0.498 |
| xlmr-large | 0.412 | $0.594^D$ | 0.412 | 0.904 |

| Case | DE-100 | DE-250 | DE-500 | IS |
|---|---|---|---|---|
| mbert | 0.370 | 0.470 | 0.541 | 0.323 |
| xlmr-large | 0.330 | $0.610^D$ | $0.430^D$ | 0.289 |

Table 5: Discriminative context as labels.

systems, the less-represented labels (for example *genitive* in German *case*) are not well represented in the models, and are thus under-predicted.

**Variants**   We explored using discriminative context words for gender and case to have a more common-language label vocabulary: By associating the target word with a context word indicative of gender/case, we aim at directing the model's attention to relevant context in general while not impeding the prediction with presumably insufficiently covered terminology. For gender, we used personal pronouns in nominative case, assuming they provide useful content, e.g. when referring to the target noun. For case, we use definite (German) and indefinite (Icelandic) articles as labels. However, articles are highly syncretic (cf. Table 10), and thus not always distinct. In general, we observed some improvements for gender, and even for case (cf. Table 5), which is nevertheless still far from good. Interestingly, *xlmr-large* tends to produce lower results than *mbert* for German gender, in contrast to the results from using English labels.

This result indicates that abstract grammatical terminology is not necessarily the best and only way to address linguistic features. However, one should also bear in mind that the formulation of prompts has been shown to be finicky (Webson

---

[3]While most systems showed consistent results for the two runs, some had large differences (marked with 'D' for a difference > 0.2. This concerned to a large part Icelandic systems, and some of the German system based on *xlmr-large*.

[4]Number for German nouns is not lexicalized (unlike Spanish). Singular and plural can be the same, cf. Table 9.

and Pavlick, 2022), and that a non-optimal prompt formulation does not always lead to a bad performance: this adds a new layer of difficulty to our task as it makes a clear interpretation difficult.

## 5.2 A Broader Variety of Linguistic Tasks

Here, we study linguistic tasks that go beyond querying for standard morphological features.

**Subject-Object** This task consists in deciding whether a word is the subject or direct object of a verb in a given sentence context. We only use nouns, no pronouns, as target words. While this is similar to predicting case, this variant is easier with only the two labels *subject* and *object*, which have the additional benefit of being more commonly used than the actual labels for the different grammatical cases. In contrast to the previous experiments, we designed the set to have approximately balanced classes (see Table 11 for the label distribution). Furthermore, as many nouns can take both the subject and the object position, the same verb-noun pairs can occur several times across the data sets. The same applies in general for the three further syntactic tasks, as the realization of the respective features depends on the context.

**Particles** German verb particles can occur separately from the verb, for example *ausschneiden ↔ schneidet ... aus* (*to cut out*). There can be a large distance between particle and verb, making this task not trivial. As contrastive examples, we used sentences where a preposition (restricted to those that share the same form as frequent particles) is attached to a verb. The queries are formulated as variants of "*Is P a particle of V?*".

There are many verbs that can occur with and without a particle, as shown in the example below:

... er **spielte** noch in drei Tonfilmen **mit** ...        YES
... *he played in three sound movies particle* ...

... **spielte** die Band Konzerte **mit** Bands wie ...        NO
... *played the band concerts with bands as* ...

**PP-attachment** For this task, we select sentences with (at least) two verbs, and ask whether a prepositional phrase PP is attached to verb V, which is either the verb the PP is attached to, or a randomly selected, other verb. Due to German sentence order, there can be a long distance between the verb and an attached PP, making this task very challenging. The prompts are formulated as variants of "*Is P attached to V?*". We show an example below, where the PP *auf Niveau* is attached to the verb *spielen* (*to play*):

| Subj/Obj | DE$_{100}$ | DE$_{250}$ | DE$_{500}$ | IS | ES |
|----------|-----------|-----------|-----------|------|------|
| mbert | 0.635 | 0.736 | 0.817 | 0.752 | 0.692 |
| xlmr-l | 0.785 | 0.880 | 0.709$^D$ | 0.832 | 0.739 |

Table 6: Distinction of subject and direct object.

| German | verb particles | pp-attachm. | passive |
|--------|---------------|-------------|---------|
| mbert | 0.80 | 0.522 | 0.674 |
| xlmr-large | 0.89 | 0.501 | 0.692 |

Table 7: Further linguistic tasks for German.

... wurde den Spitzenclubs ... die Möglichkeit **geboten**, **auf** höherem **Niveau** als ... gegen die besten Clubs ... zu **spielen**.
*... were the top clubs ... the opportunity **offered at** a higher **level** than ... against the best clubs ... to **play**.*

verb=*geboten*, pp=*auf Niveau*        NO
verb=*spielen*, pp=*auf Niveau*        YES

**Passive Voice** The verb *werden* can occur in several functions, for example as an auxiliary for passive voice or future tense, see the examples below:

Der Reaktor muß für eine Woche abgestellt **werden**        YES
*The reactor must **be** turned off for a week*

... **wird** es in künftigen Fällen nicht mehr ausreichen        NO
... ***will** not be enough in future cases*

daß er ... Mitglied der Republikaner **geworden** ist        NO
*that ... he **became** a member of the Republicans*

To identify passive structures, the queries are formulated as '*Is "werden" a passive auxiliary?*'.

Table 6 shows a reasonable performance for the task of subject-object identification task across languages; as before, we observe that increasing the number of training examples leads to improvement at least for *mbert*. The other three tasks are carried out only for German; the results are showed in Table 7. We observe a mixed performance, indicating that PET can be applied to some linguistic tasks.

# 6 Conclusion

We presented a study on using prompting to access linguistic information in pre-trained LMs for German, Icelandic and Spanish. We evaluated predicting a set of morphological features, as well as a broader variety of linguistic tasks. Our results indicate that some features are indeed accessible, whereas the approach fails for features such as the complex and abstract grammatical case.

Using natural language to obtain linguistic information is potentially useful in many scenarios; querying for a variety of non-standard linguistic tasks might be an especially interesting use case, in particular for low-resource languages.

## Limitations

The work reported in this paper, a study on accessing linguistic information in pre-trained language models through natural language instructions, presents an initial but targeted study on German, Icelandic and Spanish data.

An obvious limitation lies in basing the experiments on three European languages that are not representative of many different language families. We chose German as our main target language because it has a complex enough morphology to provide a set of challenging tasks, but at the same time has excellent linguistic tools available to be used in the analysis. We furthermore chose Icelandic as a related, but lower-resourced language and Spanish with its comparatively simple nominal morphology as a contrasting language. Similarly, we tried to model features from different linguistic levels (both morphological and syntactic features) in order to provide a reasonably comprehensive overview. Exploring more language families and their respective grammatical features, as well as applying the methodology to low-resource languages makes for an interesting project that is, however, beyond the scope of this work.

We primarily focused on *accessing* information through natural-language instructions, but without actual knowledge whether the queried information is really encoded in the model. To extract linguistic information, one needs to address two questions, namely, "Is the information there at all?" and, "How can we access it?" In this study, we skip the first question but just assume that the information is available. While this is generally a reasonable assumption, as previous work has shown that there *is* knowledge about grammar and linguistic structure in LLMs, we do not know for sure whether information about the feature we are interested in is encoded in a way that corresponds to the query and the pre-defined label vocabulary.

With view to the comparatively good results of some linguistic features, we might say that information about those feature is there and can be obtained through natural-language prompts. For other features, we cannot really draw a conclusion. For example, consider grammatical case, where the prediction model clearly failed, but we saw evidence of awareness for related features in the task of distinguishing between subject and object. We thus cannot say for sure whether there is no knowledge about case in terms of the labels we used, or

whether it is just not accessible by means of our current method. We plan to work on separating these distinctions in the future.

## Ethics Statement

The authors see no ethical concerns with the work presented in this paper.

## Acknowledgements

This publication was supported by LMUexcellent, funded by the Federal Ministry of Education and Research (BMBF) and the Free State of Bavaria under the Excellence Strategy of the Federal Government and the Länder; and by the German Research Foundation (DFG; grant FR 2829/4-1).

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

## A  Data Information and Test Set Creation

We used the UD Treebank corpora *de_gsd-ud-(dev/test/train)* (McDonald et al., 2013) for German, *es_ancora-ud-(dev/test/train)* (Taulé et al., 2008) for Spanish and *is_modern-ud-(dev/test/train)* for Icelandic. Table 8 gives the amount of training data for all three languages.

|  | German | Spanish | Icelandic |
|---|---|---|---|
| selected from (test/dev/train) | 385/186/1941 | 932/696/3118 | 197/114/606 |
| final test | 826* | 932 | 197 |
| final dev | 184 | 696 | 114 |
| final train | 100/250/500 | 100 | 100 |
| unlabeled data | 1000 | 1000 | 500 |

Table 8: Overview of the training data: number of sentences. (*: Some of the unused training data was added to the test data.)

## B  Background: German Grammar

Tables 9 and 10 provide examples for syncretism in definite articles and nouns.

| case | singular | plural |
|---|---|---|
| **nominative** | der Elefant | die Elefanten |
| **accusative** | den Elefanten | die Elefanten |
| **dative** | dem Elefanten | den Elefanten |
| **genitive** | des Elefanten | der Elefanten |

Table 9: Inflected forms of *Elefant* (*elephant*): there is only one distinct singular form (nominative case), whereas all other forms are identical with the plural form.
(cf. www.duden.de/deklination/substantive/Elefant)

|  | MASC | FEM | NEUT | MASC | FEM | NEUT |
|---|---|---|---|---|---|---|
| NOM | **der** | die | das | die | die | die |
| ACC | **den** | die | das | die | die | die |
| DAT | **dem** | der | dem | den | den | den |
| GEN | **des** | der | des | der | der | der |
|  | SINGULAR | | | PLURAL | | |

Table 10: Definite articles: bold-faced are used as labels for the prediction of *case* (der, den, dem, des).

## C  Further Linguistic Tasks: Data Details

Table 11 shows the label distribution for the subject-object distinction task. Table 12 shows the label distribution for the three syntactically inspired tasks.

## D  Alternative Approach: Probing

As an ablation, we trained linear classifier probing models for each morphological feature. The

|  | DE | IS | ES |
|---|---|---|---|
| **train** | object: 49 | object: 52 | object: 45 |
|  | subject: 51 | subject: 48 | subject: 55 |
| **test** | object: 505 | object: 408 | object: 500 |
|  | subject: 495 | subject: 402 | subject: 500 |

Table 11: Overview of the label distribution of the subject-object experiment.

|  | Particles | PP attachm. | Passive |
|---|---|---|---|
| **train** | part-yes: 54 | attach-yes: 40 | pass-yes: 50 |
|  | part-no: 46 | attach-no: 60 | pass-no: 50 |
| **test** | part-yes: 160 | attach-yes: 504 | pass-yes: 224 |
|  | part-no: 140 | attach-no: 496 | pass-no: 217 |

Table 12: Overview of the label distribution of the German syntactic tasks.

training conditions between PET and the probing models are not comparable in a straightforward way. The probing models are based on 100 training sentences (PET training data) and the additional 1000 sentences used in the distillation step (1100 sentences total). We observe that for most features, at least one variant of PET is better than the respective probing result, with the exception of case, for which PET is always worse. This shows that prompting is generally well-suited to accessing linguistic information, and is in some cases even superior to prompting, but also that some features are out-of-reach with this method. See Tables 13 and 14 for detailed results.

To investigate how PET fine-tuning changes the models, we also probed the resulting PET models. Overall, the PET models tend to perform worse on the probing task, except for those that obtain very high results in the morphological prediction task. It seems that PET training does not add (much) information relevant for probing, and that less well-performing PET models impair the original model with regard to the probing task. See Table 15 for detailed results.

For each morphological feature, we trained a separate probing model based on a linear classifier, running over 25 epochs. In order to be comparable to the training conditions of PET, the training data consists of the labels of the respective feature for nouns (number, gender or case) and verbs (number, tense), whereas all other words in the sentence are labelled NA.

As the comparison between the training conditions of PET (100 labeled training examples in addition to 1000 unlabeled examples to train the final

| | | DE | IS | ES |
|---|---|---|---|---|
| **verb** | mbert | $0.814^P$ | 0.766 | $0.864^P$ |
| **tense** | xlmr-large | $0.831^P$ | 0.761 | $0.838^P$ |
| **verb** | mbert | 0.895 | 0.797 | $0.902^P$ |
| **number** | xlmr-large | 0.823 | $0.766^P$ | $0.866^P$ |
| **noun** | mbert | $0.834^P$ | $0.792^P$ | $0.856^P$ |
| **number** | xlmr-large | $0.752^P$ | 0.761 | $0.791^P$ |
| **noun** | mbert | $0.677^P$ | 0.584 | 0.829 |
| **gender** | xlmr-large | $0.551^P$ | $0.65^P$ | $0.714^P$ |
| **noun** | mbert | 0.429 | 0.731 | – |
| **case** | xlmr-large | 0.368 | 0.635 | – |

Table 13: Probing experiment with n=100 sentences training data (corresponding to the n=100 labeled examples in the PET experiment). $^P$: the respective PET-100 model (EN labels or discriminative context), in Tables 4 or 5, is better.

| | | DE | IS | ES |
|---|---|---|---|---|
| **verb** | mbert | 0.891 | 0.812 | $0.912^P$ |
| **tense** | xlmr-large | $0.89^P$ | 0.792 | $0.938^P$ |
| **verb** | mbert | 0.92 | 0.817 | $0.954^P$ |
| **number** | xlmr-large | 0.915 | $0.832^P$ | $0.959^P$ |
| **noun** | mbert | $0.87^P$ | 0.817 | $0.916^P$ |
| **number** | xlmr-large | $0.861^P$ | 0.868 | $0.87^P$ |
| **noun** | mbert | $0.723^P$ | 0.629 | 0.879 |
| **gender** | xlmr-large | $0.755^P$ | $0.746^P$ | $0.813^P$ |
| **noun** | mbert | 0.643 | 0.766 | – |
| **case** | xlmr-large | 0.682 | 0.802 | – |

Table 14: Probing experiment with n=1100 sentences training data (n=600 for IS), corresponding to the 100 sentences for PET training and the additional sentences used for the distillation step. $^P$: the respective PET-100 model (EN labels or discriminative context), in Tables 4 or 5, is better.

| | | DE | IS | ES |
|---|---|---|---|---|
| **verb** | PET mbert | 0.860 | 0.736 | *0.926 |
| **tense** | PET xlmr-large | 0.874 | 0.777 | *0.954 |
| **verb** | PET mbert | 0.866 | 0.812 | 0.947 |
| **number** | PET xlmr-large | 0.910 | *0.853 | 0.951 |
| **noun** | PET mbert | 0.852 | 0.726 | 0.912 |
| **number** | PET xlmr-large | 0.849 | 0.838 | *0.872 |
| **noun** | PET mbert | *0.729 | 0.624 | 0.821 |
| **gender** | PET xlmr-large | *0.780 | 0.746 | *0.865 |
| **noun** | PET mbert | 0.609 | 0.665 | – |
| **case** | PET xlmr-large | 0.630 | *0.807 | – |

Table 15: Probing experiment with n=1100 sentences (600 for IS) training data applied to the respective best PET-100 model (EN labels or discriminative context). *: PET model is better than the respective original model in Table 14.

# E   Computational Resources

We use the implementation of PET available from https://github.com/timoschick/pet.

We ran the PET experiments on CPU with up to 20 threads (*Intel(R) Xeon(R) CPU E5-2630 v4*). The time used per experiment depends on the particular setting (mainly number of patterns) and the model size. For example, training the model for German noun number with n=100 training sentences on three patterns took approximately 2 hours for mbert and 6.5 hours for xlmr-large.

classifier) and the probing models is not straightforward, the probing models are based on 100 training sentences (PET training data), as well as on 1100 training sentences (PET training data + 1000 extra sentences), with the respective feature labels given on nouns and finite verbs. This provides more training data to the probing models, as PET only sees the label for one noun or verb per sentence in the training data, and also does not know the labels of the data for the distillation step.

We evaluate the probing models on the same target words as we do PET, counting prediction accuracy on those words while ignoring predictions for other words in the sentence.