# OpenReview forum: "A Study on Accessing Linguistic Information in Pre-Trained Language Models by Using Prompts"
_EMNLP/2023/Conference — EMNLP 2023 Main_

### Official Review · Reviewer_uGEU · 2023-07-26

**Soundness:** 3

**Excitement:**

3: Ambivalent: It has merits (e.g., it reports state-of-the-art results, the idea is nice), but there are key weaknesses (e.g., it describes incremental work), and it can significantly benefit from another round of revision. However, I won't object to accepting it if my co-reviewers champion it.

**Paper Topic And Main Contributions:**

This work studies the ability to access linguistic information by various LMs via prompting. The authors use PET as methodology and experiment with German, Icelandic, and Spanish data from UDT. They test morphological and syntactical features, framing the task as classification and formulating prompts asking for, e.g., the grammatical case of a word-form. The results show that some features are more easily accessible than others, with substantial differences among different languages.

**Questions For The Authors:**

- A: 4-5, 23-24: you say that there is no easy method to directly obtain linguistic information from LMs. Can you please substantiate this statement (maybe by citing some work)?
- B: In Table 1, you give the English equivalent of the sentence given as context. It is not entirely clear whether you are using it here simply as a gloss, or you are giving that to the model too. I guess it is the former (and if so, clarify that it is just a gloss).
- C: 146-147: "general-language discriminative context"; what does it mean? And, can you provide an example?
- D: Sec. 5: why do also study probing? Is it relevant/comparable to PET? Do you need Sec. 5 at all?
- E: Are you going to release the exact data you used?

**Reasons To Accept:**

- The use of PET helps in mitigating some problems that may arise with prompting.
- The work has a clear scope and the Limitations are explicitly defined.

**Reasons To Reject:**

- I am not entirely clear with the terminology used. For example, what is the difference between "tangible" and "abstract" features? Where do these terms come from? Did you come up with them? If so, where do you draw the line? If not, please cite a work where these terms are defined.
- The previous point affects the clarity of the manuscript's exposition. For example, in line 196, the term "tangible" is used again with a dubious definition, and "complex and abstract" is used in line 328. In addition, other points would need rephrasing. For example: line 209 "common-language label vocabulary" and the subsequent paragraph are unclear to me.
- Sec. 4.2 is "rushed" with many details that are crucial for understanding better what was done relegated to App. C.
- I would like to see a conclusive discussion clearly summarizing the results obtained.

**Reproducibility:**

3: Could reproduce the results with some difficulty. The settings of parameters are underspecified or subjectively determined; the training/evaluation data are not widely available.

**Reviewer Confidence:**

4: Quite sure. I tried to check the important points carefully. It's unlikely, though conceivable, that I missed something that should affect my ratings.

**Typos Grammar Style And Presentation Improvements:**

- It would be advisable to move Related Work after Introduction to give the unaccustomed reader a comprehensive background.
- Line 185: 'T'able 7 (check all instances).
- Note 2: open inverted quote.
- All links are not clickable.
- All tables could be more appealing (e.g., changing fonts, using booktabs, etc.)

---

> ### Author Rebuttal · Authors · 2023-08-28
>
> Thank you for your thoughtful review. We would like to submit that the main points you raise are all writing and terminology issues and can be easily resolved for a camera-ready. Of course, we will do our best to make the terminology more consistent and clear.
> As we said above, we will keep a close eye on what details can and should move up from the Appendix when revising.
>
> To answer your questions:
>
> A) Question about “no easy method to obtain linguistic information” in the introduction:
> This refers to obtaining information in an intuitive way, e.g. through natural language queries, that is also flexible to handle different types of (language-specific) linguistic information. We will make this more precise.
>
> B) Question about gloss in table 1:
> Yes, this is a gloss, indicated by the italics. We can make this more explicit.
>
> C) Question about “general-language discriminative context”:
> This refers to the experiments in Table 3, where we don’t use the actual labels (such as nominative or feminine) for case and gender, but a general-language context word indicative of the feature (We use articles and pronouns, cf. lines 214 ff. For example, the definite article “dem” only occurs in a Dative context in German, and we thus use it as a label instead of the word “Dative”).
> We will make sure to explain this better already in Section 3.
>
> D) Question about probing:
> Funnily enough, this was requested in a previous round of reviews, as traditional probing is an existing, well-known method to access morphosyntactic information. As well, reviewers worried that the pattern-exploiting training was significantly changing the information available in the model. We will clarify the motivations for this.
>
> E) Question about releasing datasets:
> Yes, we can release the data sets we used.
>
> Thank you as well for the typos you caught!

---

### Official Review · Reviewer_CAHM · 2023-08-02

**Soundness:** 4

**Excitement:**

4: Strong: This paper deepens the understanding of some phenomenon or lowers the barriers to an existing research direction.

**Missing References:**

Nothing to report, except that the first reference of the list (Blevins et al., 2023) seems to be incomplete, i.e. some details probably missing.

**Paper Topic And Main Contributions:**

The paper addresses the very interesting, challenging and relevant question whether linguistic information in pretrained multilingual language models can be accessed through human language. The experiments reported cover German, Icelandic and Spanish. The manuscript addresses an original and timely issue and given its limited length (4 pages of actual paper) provides an interesting and well-argued discussion of the topic.

**Questions For The Authors:**

No specific questions, but below I list some comments and suggestions that the author(s) might consider when revising the paper.

**Reasons To Accept:**

Interesting issue that is addressed convincingly with results that are discussed clearly and correctly. Sound and valid methodology. Good discussion of the limitations of the study in the dedicated section (lines 335-394). The motivation and explanation for the interesting choice of languages covered in the paper (i.e. German, Icelandic and Spanish) given in lines 150-162 is very good and convincing.

**Reasons To Reject:**

None particularly that is worth mentioning or that struck me particularly negatively.

**Reproducibility:**

4: Could mostly reproduce the results, but there may be some variation because of sample variance or minor variations in their interpretation of the protocol or method.

**Reviewer Confidence:**

4: Quite sure. I tried to check the important points carefully. It's unlikely, though conceivable, that I missed something that should affect my ratings.

**Typos Grammar Style And Presentation Improvements:**

The paper is well-structured and generally well-written, it was generally a pleasure to read.

Some comments:

- The German of this reviewer is a bit rusty, but I'm fairly sure that "(made vs. has made)" in line 553 should instead be "(made vs. (I) have made)", as a gloss for "e.g. machte vs. habe gemacht" (with italicised words where appropriate, as is done in the manuscript), on the grounds that the auxiliary verb in the compound tense "habe gemacht" in lines 552-553 is first person singular. For this reason, I believe that the English gloss is incorrect here, please double-check and amend if/as required.

- Line 555: "(Past: preterite tenses)" should probably be "(Past: preterite tense)", in the singular, not plural form (italicised words are fine).

- I found it odd that section 6 on related work (lines 286-319) is located towards the very end of the paper, just before the conclusion. Unless there are requirements specifically against this, I would suggest to move it earlier on in the paper as customary, e.g. after the Introduction section. Also, this section on related work could highlight better and more clearly the specific contributions of the manuscript under review.

- The table numbers (and the sections A / B / C etc.) are not mentioned in the right sequence in the running text of the manuscript, which is quite annoying (and has no specific reason, I believe, unless I missed something?).

- In Section 1 (introduction), paragraph with lines 067-074, the author(s) could formulate those points as research questions and hypotheses that guided their study, rather than as a pre-empted discussion of the findings. In addition, in a similar vein the author(s) could discuss the potential applications of their research in a bit more detail.

- What the author(s) call(s) "section A / B / C" etc. seem more appendices to me. In addition, sections (or possibly appendices) A, B, C and D (to a lesser extent section/appendix E) seem integral part of the paper, insofar as they report data and results of the experiments that were conducted for the study, and for this reason should probably be actual parts/sections of the main body of a long(er) paper, rather than "left-overs" lumped together at the end of the manuscript. But I understand that this would probably change the submission type of the manuscript, also for the purposes of its refereeing and evaluation for acceptance...

---

> ### Author Rebuttal · Authors · 2023-08-28
>
> Thank you for your kind review and helpful notes. We will address them as best we can in the camera-ready.
> You are right that moving all or most of the appendices to the body would change the submission type. As we noted in another rebuttal, we do believe that the most important information is included in the paper body, but we will keep a close eye on what details can and should move up when revising.

---

### Official Review · Reviewer_SkRy · 2023-08-04

**Typos Grammar Style And Presentation Improvements:** 1. Line 241
**Soundness:** 3

**Excitement:**

4: Strong: This paper deepens the understanding of some phenomenon or lowers the barriers to an existing research direction.

**Missing References:**

I didn't note any missing references.

**Paper Topic And Main Contributions:**

This paper tries to determine if linguistic information about words and sentences can be accurately obtained from LLM's with prompts that directly request it. The paper describes experiments that test this question based on Icelandic, German and Spanish, which represent some diversity in morphology. It shows that some information is available from LLM's at a modest level of accuracy.

**Questions For The Authors:**

A. The paper should explain why the authors chose to mix the language of the prompt with the language of the queried feature. This seems like it would introduce some complexities, for example (as the paper states) English labels referring to another language's grammar.

B. For the "subject/object" discrimination task, the paper should clarify whether these were the only choices in a given prompt and that one of the choices was correct (that is, no prepositional objects or nouns in compound noun expressions).

C. What does "best patterns" mean at line 544? "Using the same questions, formulated in English, for all languages, we used the dev set to select the 3 best patterns out of 5 initially given patterns assuming the respective patterns lead to different outcomes across languages."

D. Line 318 "We are also interested in code-switched prompts." This is the first time the paper refers to code-switching. Is that the same as "mixed-language" prompts? If not, it should be explained. If so, the terminology should be kept consistent.

E. Why are the Icelandic results in Table 2 so much worse than the German results? The Icelandic results are almost at chance. This should be discussed.

**Reasons To Accept:**

Although as reviewed in section 6, "Related work", there seems to be quite a bit of previous work in attempting to extract grammatical information from LLM's, with similar goals to those in this paper, this paper presents results on an interesting variety of grammatical features and compares the results for three non-English languages. For this reason, the results are interesting and could be valuable to the EMNLP community.

**Reasons To Reject:**

Because the relative frequency of each of the items being distinguished can be quite different, (Table 7), the paper should also present results as precision, recall, and F1, in addition to accuracy, in order to clarify the meaning of the results because of the unequal numbers of each category in the data.

There is a lot of information included in the Appendices which I found necessary to read in order to evaluate the work. For example, Table 7 "label distribution of features per language" is very important for understanding the meaning of the accuracy results. Another example is the results for the probing task (Section 5) can't really be evaluated without referring to Tables 13 and 14 in the Appendix. In Section 5 the approaches being compared (PET vs. probing) are just described as being better or worse. The quantitative results are only provided in the Appendix. To compare the PET method to probing, there should ideally be a chart comparing the results in Table 2 to those in Table 13.  Another important piece of information found only in the Appendix is the definition of "tense". The main paper does not provide a definition of tense -- it is necessary to read Appendix A to find out what "tense" means in this experiment.

**Reproducibility:**

4: Could mostly reproduce the results, but there may be some variation because of sample variance or minor variations in their interpretation of the protocol or method.

**Reviewer Confidence:**

4: Quite sure. I tried to check the important points carefully. It's unlikely, though conceivable, that I missed something that should affect my ratings.

---

> ### Author Rebuttal · Authors · 2023-08-28
>
> Thank you for your thoughtful review. You are right that precision/recall/F1 scores would be helpful. As they were not output by the PET tool, we did not initially think to calculate these, but we can add this for the camera-ready based on the saved outputs. Yes, we moved a lot of details to the Appendices due to space constraints, and we will evaluate closely which points can or need to move up for the camera-ready. However, we believe that the main story is indeed contained in, and can be understood from, the body of the paper. Any missing elements will be straightforward to fix in the writing.
>
> In response to your questions:
>
> A) Question about mixed-language prompts:
> It is true that mixed-language queries add a certain complexity. However, it allows to keep the setting the same when testing several languages. Furthermore, many features have the same general meaning across languages (such as singular vs. plural) and might be better represented in English due to the larger amount of PLM training data. For other features, this is less clear, which is why we also investigate discriminative context as labels in the target language (cf. Table 3).
>
> B) Question about labels in the subject-object task:
> The task is formulated with the two labels “subject” and “object”. And yes, one of the labels always applies.
>
> C) Question about best patterns:
> This refers to the best-performing patterns on the development set.
>
> D) “Code-switched” vs. “mixed language”:
> Yes, “code-switched” here refers to the prompts mixing languages. We will adjust the terminology.
>
> E) Performance of Icelandic:
> Icelandic has less training data overall in our data set, and the percentage of Icelandic in the PLM training data is much smaller than that for German and Spanish, (cf. https://data.statmt.org/cc-100/). We assume that this is the main factor for the comparatively bad outcome of Icelandic.

---

### Meta-Review · Area_Chair_kdck · 2023-09-19

**Recommendation:** 4

**Metareview:**

The authors develop prompts to shed light on LLM’s access to explicit grammar principles. The authors evaluate on Icelandic, German, and Spanish. All reviewers appreciated the motivation of this work, but noted that it could benefit from clarifying its exposition.

---

### Decision · Program_Chairs · 2023-10-07

**Decision:**

Accept-Main

**Comment:**

The authors develop prompts to shed light on LLM’s access to explicit grammar principles. The authors evaluate on Icelandic, German, and Spanish. All reviewers appreciated the motivation of this work, but noted that it could benefit from clarifying its exposition.